# Dynasore Protects Corneal Epithelial Cells Subjected to Hyperosmolar Stress in an In Vitro Model of Dry Eye Epitheliopathy

**DOI:** 10.3390/ijms24054754

**Published:** 2023-03-01

**Authors:** Rafael Martinez-Carrasco, M. Elizabeth Fini

**Affiliations:** 1New England Eye Center, Tufts Medical Center and Department of Ophthalmology, Tufts University School of Medicine, Boston, MA 02111, USA; 2Graduate School of Biomedical Sciences, Tufts University, Boston, MA 02111, USA

**Keywords:** corneal epithelium, ocular surface, epitheliopathy, dry eye, hyperosmolar stress, oxidative stress, endoplasmic reticulum stress, unfolded protein response, dynasore

## Abstract

Epitheliopathy at the ocular surface is a defining sign of dry eye disease, a common disorder that affects 10% to 30% of the world’s population. Hyperosmolarity of the tear film is one of the main drivers of pathology, with subsequent endoplasmic reticulum (ER) stress, the resulting unfolded protein response (UPR), and caspase-3 activation implicated in the pathway to programmed cell death. Dynasore, is a small molecule inhibitor of dynamin GTPases that has shown therapeutic effects in a variety of disease models involving oxidative stress. Recently we showed that dynasore protects corneal epithelial cells exposed to the oxidant tBHP, by selective reduction in expression of CHOP, a marker of the UPR PERK branch. Here we investigated the capacity of dynasore to protect corneal epithelial cells subjected to hyperosmotic stress (HOS). Similar to dynasore’s capacity to protect against tBHP exposure, dynasore inhibits the cell death pathway triggered by HOS, protecting against ER stress and maintaining a homeostatic level of UPR activity. However, unlike with tBHP exposure, UPR activation due to HOS is independent of PERK and mostly driven by the UPR IRE1 branch. Our results demonstrate the role of the UPR in HOS-driven damage, and the potential of dynasore as a treatment to prevent dry eye epitheliopathy.

## 1. Introduction

The wet mucosal epithelia of the cornea and conjunctiva are directly exposed to a hostile environment, with a desiccating atmosphere and a plethora of potential damaging agents. The tear film provides the ocular surface epithelia with the necessary aqueous milieu, as well as microbicidal, anti-inflammatory, and growth factors [1]. A variety of factors, including aging and ocular conditions, can alter the tear composition, causing epitheliopathy [2]. One of these is a common condition known as dry eye disease (DED), that affects 10% to 30% of the population worldwide [3,4]. Despite the significant impact of dry eye on vision and quality of life, there are only a few FDA-approved therapies for the treatment of dry eye, and most of them target the inflammatory component. Corticosteroids, such as the recently approved Loteprednol, can only be used for a short period, due to the risk of adverse effects [5]. The other two anti-inflammatory FDA-approved drugs—cyclosporine A and lifitegrast—improve symptoms in only a subset of patients [6], and both showed limited evidence of effectiveness against epitheliopathy in human clinical trials [7,8,9,10]. Thus, there continues to be an unmet need for new therapeutics.

Hyperosmolarity of the tear film is one of the main drivers of DED [11]. Hyperosmolar stress (HOS) triggers a series of cell responses that act to compensate for the imbalance of water and electrolytes [12]. It has been widely reported that the signaling pathways in epithelia activated in response to HOS, can end in the expression of inflammatory mediators and cell death [13]. Cytokines, matrix metalloproteinases, and cell death products are then released to the extracellular medium and tears, contributing to creating or sustaining a proinflammatory environment at the ocular surface, that leads to epitheliopathy [14,15].

Recent studies have evidenced endoplasmic reticulum (ER) stress as a mediator in DED, that contributes to epitheliopathy at the ocular surface [16,17,18,19]. ER stress occurs when the protein folding capacity of the ER is compromised, either by an excessive load of translated proteins or a defect in molecular chaperones [20]. The cell responds by activation of the unfolded protein response (UPR), a complex cellular process that alleviates the protein load of the ER, while increasing the expression of molecular chaperones. The UPR is activated by three sensors situated at the ER membrane: activating transcription factor 6, or ATF6; inositol-requiring enzyme 1, or IRE1; and eukaryotic translation initiation factor 2 alpha kinase 3, also known as PKR-like ER kinase, or PERK. The activated, phosphorylated IRE1 splices a short sequence from the mRNA for X-box-binding protein 1 or XBP1, causing a translation frameshift. The resultant spliced form of XBP1 mRNA (sXBP1) encodes a potent transcription factor, whose target genes are directed to eliminate unfolded proteins from the ER and produce more chaperones. When activated, also by phosphorylation, PERK can, in turn, phosphorylate eukaryotic translation initiation factor 2 subunit alpha, or eIF2α subunit, with a reduction in general translation as the most relevant effect. If ER stress continues unresolved, programmed cell death is orchestrated by overexpression of C/EBP-homologous protein 10 or CHOP [21,22].

We recently reported that dynasore protects corneal epithelial cells in culture, and the mouse corneal epithelium ex vivo, against the damaging effects of the oxidant tert-butyl hydroperoxide (tBHP) [23,24]. Dynasore is a small molecule selected for its ability to inhibit endocytosis by impairing the GTPase activity of classic dynamins [25]. Since then, different studies have demonstrated cytoprotective roles for dynasore [26,27,28], and its potential as a therapeutic for a variety of diseases [29,30,31,32,33,34]. We found that tBHP exposure activates the UPR in corneal epithelial cells in culture, and that dynasore selectively reduces PERK branch activation, and consequently, also reduces the expression of CHOP [23].

In the present study, we investigated the protective effects of dynasore against HOS.

## 2. Results

### 2.1. Dynasore Protects against HOS-Induced Cell Death

Oxidative stress, mediated by excessive accumulation of reactive oxygen species (ROS), plays a prominent role in mucosal epitheliopathy from a variety of causes [35,36,37,38]. In our previous study, we demonstrated that dynasore prevents corneal epithelial cell death due to tBHP exposure [23]. Since HOS caused by dysfunctional tears in DED also induces oxidative stress and corneal epitheliopathy [39], we hypothesized that dynasore might also be protective against HOS. To investigate this idea, we again used a telomerase-immortalized human corneal limbal epithelial (HCLE) cell line [40]. We analyzed cell viability and cell death in monolayer cultures of HCLE cells subjected to HOS using an assay that employs calcein-AM dye to identify live cells, and propidium iodide (PI) dye to identify dead cells. Representative results are shown in Figure 1. In unstressed cultures, almost all cells were viable, as indicated by the calcein-AM+, PI−staining pattern. However, cell death was induced by HOS, as evidenced by a dramatic increase in the percentage of calcein-AM−, PI+ stained cells. Significantly, dynasore treatment inhibited the loss of calcein-AM staining due to HOS, while also reducing the increase in PI+ cells.

Previous studies have demonstrated the activation of caspase-3 in relation to HOS-induced cell death [41,42,43,44]. Therefore, we further investigated the effects of dynasore treatment on caspase-3 activation in monolayer HCLE cell cultures. Representative results are shown in Figure 2. We used a fluorescence-based assay for caspase-3, which allowed the visualization of enzyme activation under different conditions. Using this assay, we confirmed the increase in caspase-3 activity in cells subjected to HOS, and we demonstrated its inhibition by dynasore, for the first time.

### 2.2. Dynasore Reduces HOS-Induced Inflammasome Activation

Stratified HCLE cell cultures with mucosal differentiation, are a better model of the corneal epithelium in vivo (reviewed in [45]). We further tested the protective effects of dynasore against HOS in these cultures. In these experiments, we used a third cell death assay that measures plasma membrane integrity, as evidenced by release of lactate dehydrogenase (LDH) activity into the cell culture medium. Representative results are shown in Figure 3A. HOS increased plasma membrane permeability, as demonstrated by the increase in LDH in the cell culture medium, and treatment with dynasore was protective.

One of the effects of HOS that contributes to epitheliopathy is the induction of the NOD-, LRR- and pyrin domain-containing protein 3 (NLRP3) inflammasome [46]. The NLRP3 inflammasome is a protein complex constituted by a sensor (NLRP3), an adaptor (apoptosis-associated speck-like protein containing a CARD; abbreviated ASC), and an effector (caspase-1). This protein complex is responsible for the activation of cytokines and the formation of membrane pores that permit the release of these inflammatory mediators. Using stratified HCLE cell cultures, we investigated the effects of dynasore on caspase-1 activity with a luciferase-based assay. Representative results are shown in Figure 3B. The results confirmed that HOS induced an increase in caspase-1 activity, and that this was abrogated in the presence of dynasore. We further analyzed the gene expression of *NLRP3*, and gasdermin-D (*GSDMD*), the final effector of inflammasome-induced pore formation and cell death. Representative results are shown in Figure 3C. The transcript levels of both genes were elevated in HOS-treated cells, but not when dynasore was present.

### 2.3. Dynasore Regulates the Effects of UPR Activation

In our previous work [23], we showed that tBHP exposure of both monolayer and stratified HCLE cells induces ER stress and activates the UPR. Here we investigated dynasore’s capacity to protect HCLE cells exposed to tunicamycin, an inhibitor of N-glycosylation that causes ER stress [17]. Representative results are shown in Figure 4. Tunicamycin treatment for 24 h induced a significant increase in released LDH activity. However, when cells were exposed to tunicamycin in the presence of dynasore, the increase in LDH activity was prevented. These results reinforce the idea that dynasore protects cells subjected to ER stress.

Recently it was demonstrated that the UPR is also induced by HOS [41]. We confirmed this, showing that stratified HCLE cells subjected to HOS exhibited an increase in the UPR markers sXBP1 and CHOP (Appendix A). We next examined the effects of dynasore on UPR activation in cells subjected to HOS. Representative results are shown in Figure 5. As our previous study had pointed to the PERK pathway as a specific target for dynasore activity, we first focused here. Some of the effects of PERK are mediated through activating transcription factor 4 (ATF4), which is translationally upregulated by ER stress in an eIF2α phosphorylation-dependent manner [20]. To assay ATF4 expression, we transduced HCLE monolayer cells, with an ATF4 reporter construct, which expresses a nuclear-located red fluorophore when ATF4 expression is induced by phosphorylated eIF2α. Then, HCLE cells were exposed to either HOS or tunicamycin and monitored for 16 h. The tunicamycin treatment elicited an increase in the fluorescence of the ATF4 reporter. Surprisingly, we did not find an increase in fluorescence levels in cells subjected to HOS (Figure 5A). Of note, tunicamycin’s effects were counteracted by the treatment with dynasore, as well as with the PERK inhibitor GSK2606414 (Figure 5B). In order to confirm that HOS did not induce PERK activation, we examined the phosphorylation of eIF2α. Western blot analysis showed no change in P-eIF2α when HCLE cells were exposed to HOS (Figure 5C). Despite the lack of increase in P-eIF2α, its levels were slightly reduced by dynasore.

### 2.4. Dynasore Increases the Number of Cells Expressing sXBP1, but Reduces Magnitude of Response

Next, we used qPCR to quantify expression of the IRE1 branch marker *sXBP1* and the PERK branch marker *CHOP* in stratified HCLE cell cultures subjected to HOS in the presence of dynasore. An increase in *sXBP1* was observed in cells subjected to HOS alone that was not reverted in dynasore-treated cells, which actually presented a trend toward more increased *sXBP1* expression (Figure 6A). *CHOP* expression was also increased by HOS, and dynasore did not significantly reduce it.

It has been described that the IRE1/sXBP1 branch can also induce an increase in CHOP expression [47,48,49]. It was intriguing, however, to find a slight increase in sXBP1 with dynasore treatment, without that being reflected in CHOP expression. This led us to investigate sXBP1 expression at a single cell level by using also a sXBP1 reporter. This reporter expresses a nuclear-localized green fluorescent protein when IRE1 performs splicing on the sXBP1 mRNA. After 24 h, the number of sXBP1+ cells was increased among the cells exposed to HOS and tunicamycin, including those treated also with dynasore (Figure 6B,C). This increase was observed also in cells treated with dynasore only. However, the intensity of staining per cell was significantly reduced by dynasore in all groups (Figure 6C). Therefore, although dynasore did not reduce the number of cells expressing sXBP1, the levels of sXBP1 per cell remained low.

## 3. Discussion

Tear hyperosmolarity is one of the central factors causing ocular surface damage in DED [50]. Increased osmolarity in the extracellular medium is a potent stressor that induces programmed cell death [13,43]. Corneal epithelial cells exposed to high osmolarity release proteases and proinflammatory products (e.g., MMP9, TNF-a, and IL-1b) [14,15]. In the present study, we demonstrate that treatment with dynasore inhibits the programmed cell death pathways triggered by HOS. We found that dynasore can protect against ER stress and maintain homeostatic levels of UPR activity, in agreement with our previous findings [23]. In the process, we have described that UPR activation in corneal epithelial cells under HOS is PERK independent, and mostly driven by IRE1.

In recent years, inflammasomes have received increasing attention in ocular surface diseases. The NLRP3 inflammasome is activated in mice with experimentally induced dry eye [51]. Zheng and colleagues found NLRP3, caspase-1, and IL-1b increased in impression cytology samples from patients with dry eye, as well as in cultured cells exposed to hyperosmolar medium [46]. After inflammasome activation, caspase-1 can also cleave proteins of the gasdermin family, such as GSDMD, which creates pores at the plasma membrane causing cell death by pyroptosis. Pyroptosis has been demonstrated in mice with induced dry eye and cells exposed to hyperosmolar stress [52]. In agreement with this, our results show an increase in caspase-1 activity induced by HOS, as well as an increase in the expression of *NLRP3* and *GSDMD*. Furthermore, our results with calcein and PI staining, and LDH activity, are compatible with a lytic form of cell death as is induced in pyroptosis. The inhibition of these elements by dynasore suggests a potential therapeutic effect of this molecule in ocular surface disease.

The increase in NLRP3 in corneal epithelial cells can be elicited by elevated ROS, and some authors have proposed the release of mitochondrial DNA as a specific inducer of NLRP3 activation [51,53,54,55]. ER stress is also known to cause activation of NLRP3, by different mechanisms, which can involve CHOP, activation of JNK, and generation of ROS [56]. We did not explore the induction of the inflammasome by elements of the UPR, but we propose ER stress as an inducer of lytic cell death under HOS, via the inflammasome. In this case, the link between both pathways could be CHOP but, as indicated below, an overexpression of sXBP1 leads to JNK activation, which is a known inducer of NLRP3.

ER stress and the subsequent activation of the UPR have received increasing attention in ocular disease. The UPR is found to be activated at the ocular surface in patients with Sjögren syndrome and cicatricial pemphigoid [16,17,18,19]. Moreover, systemically induced ER stress in mice causes symptoms compatible with dry eye [57]. As we show here, hyperosmolarity can cause ER stress. Kidney tubular cells are frequently exposed to high osmolarity conditions, causing CHOP-mediated cell death [58]. We show here that ER stress-induced cell death after treatment with tunicamycin is reversed by dynasore. Specifically, dynasore reduced the activation of the PERK branch similarly to the PERK inhibitor GSK2606414. This agrees with our previous study, where we found that dynasore was able to reduce the activation of the PERK branch in corneal epithelial cells under oxidative stress [23]. Therefore, our results show that the effects of dynasore are not just restricted to the context of oxidative stress.

A recent study demonstrated that TUDCA, a chemical chaperone known to alleviate ER stress, effectively reduced HOS-induced damage and caspase-3 activation [41]. In this study, we show that treatment with dynasore reduces the activation of caspase-3. However, we found that cell damage caused by HOS was not linked to activation of PERK, despite observing an increase in *CHOP* expression. Although CHOP is more commonly associated with PERK activation, its expression can also be induced through the IRE1 branch, typically via JNK phosphorylation [47,48,49]. Interestingly, JNK has been previously associated with HOS-induced cell death and inflammation [14,15,43]. Although our qPCR results for *sXBP1* and *CHOP* did not show a reduction with dynasore, analysis at a single cell level with sXBP1 reporter allowed us a different perspective: while dynasore can cause an increase in the number of cells expressing sXBP1, the levels of expression per cell are reduced compared to cells without dynasore. Activation of IRE1 is necessary and beneficial for the cell, as it contributes to maintaining homeostasis. However, an excessive activation of IRE1 can be a trigger for programmed cell death and inflammasome activation [59,60]. Therefore, by keeping IRE1 activation low, dynasore can contribute to prevent cell death, while allowing the beneficial effects of UPR.

Dynasore has been proposed as a candidate therapeutic to treat diseases involving abnormal mitochondrial dynamics [33,34]. Our results presented here suggest that it might also be valuable for the treatment of ocular surface epitheliopathy. A previous study from our group showed that dynasore had no immediate effect on vital dye uptake once damage was done [24]. However, ocular surface epithelia are constantly turning over, with new cells rising up in the layers as apical cells are desquamated [61,62]. We speculate that a continuous application of dynasore would protect the new cells that rise to the surface. Then with continued topical application, apical cells would be protected from damage. The expression of ER stress markers, such as CHOP or sXBP1, could be assessed non-invasively in impression cytology samples in a future clinical test. In addition, we show in this paper a reduction in inflammasome activation after dynasore treatment. Therefore, the presence of active IL-1b and other caspase-1 cleaved cytokines released into the tears might also serve as biomarkers.

In conclusion, our results demonstrate a promising therapeutic effect for dynasore to reduce corneal epithelial damage caused by hyperosmolarity and ER stress. We propose that dynasore contributes to maintaining the homeostatic functions of the UPR, reducing the activation of programmed cell death pathways. More studies are needed to determine dynasore’s exact mechanism of action, as well as to define the role of IRE1 in the homeostasis/cell death balance in response to HOS.

## 4. Materials and Methods

### 4.1. Cell Culture

A telomerase-immortalized human corneal limbal epithelial (HCLE) cell line was used for these studies, a kind gift of Dr. Ilene Gipson, Ph.D. (Harvard Medical School). Cells were grown as previously described [40]. This cell line has been authenticated and characterized by marker expression analysis [63] and polymorphic short tandem repeat (STR) loci [64]. The cells were plated in 96- or 24-well plates at a density of 3 × 10^4^ cells/cm^2^, and grown in keratinocyte serum free medium (KSFM) (Gibco-Thermo Fisher Scientific, Waltham, Massachusetts, USA) containing 25 µg/mL bovine pituitary extract, 0.2 ng/mL epithelial growth factor (EGF), and 0.4 mM CaCl_2_.

For experiments with stratified cultures, cells were grown in 24-well plates with the described KSFM until confluence, and then switched to Dulbecco’s Modified Eagle Medium: Nutrient Mixture F-12 (DMEM/F12) medium supplemented calf serum and 10 ng/mL EGF for 7 days to induce stratification and differentiation, as previously described [40]. On the 7th day, the cells were incubated in serum-free medium for 2 h before proceeding with the treatment as indicated below. Experiments using monolayers were performed with cells at 90% confluency.

### 4.2. Cell Stress and Treatments

The cell culture medium was 312 mOsM; hyperosmolar stress (HOS) was created by adding NaCl to 69 mM to achieve 450 mOsM, as described previously [39]. To induce ER stress, tunicamycin was added instead of NaCl, to a final concentration of 10 μg/mL. Dynasore was added concomitantly to the stress agent to a final concentration of 40 μM. Stock solutions of tunicamycin and dynasore were made in dimethyl sulfoxide (DMSO), thus control cells received an equal volume of DMSO. Cells were incubated in the specified media for 24 h. Then, supernatant and cells were processed as indicated for the different assays.

### 4.3. LDH Assay

Lactate dehydrogenase activity was analyzed in cell medium using the CyQUANTTM LDH Cytotoxicity Assay (Thermo Fisher Scientific, Waltham, Massachusetts, USA). Cell medium was collected from treated cells and centrifuged at 300× *g* for 5 min to remove any cell debris. Then, 50 μL were distributed in wells of a 96-well plate in triplicate. LDH reaction mixture was prepared as indicated by the manufacturer and 50 μL were added per well. Samples were incubated for 30 min at room temperature, protected from light. The reaction was stopped by adding 50 μL of Stop Solution. Absorbance at 490 and 680 nm was measured using a Biotek Synergy H1 Microplate Reader (Winooski, VT, USA). The 680-nm absorbance value was subtracted from 490-nm absorbance to get the LDH activity value.

### 4.4. Calcein/PI Staining

Cell viability was tested by analyzing the permeability of the cells to propidium iodide (PI) and the loss of calcein staining. After the described treatments, cells were washed twice in DMEM without phenol red. Then, cells were incubated in phenol-red-free DMEM with 1 nM calcein-AM, 10 μg/mL PI, and 1 nM Hoechst 33342 dye for 30 min at 37 °C (all dyes from Thermo Fisher Scientific). Cells were observed under a Biotek Lionheart FX Automated Microscope. The percentages of calcein-negative (Hoechst+/Calcein-) and PI-positive (Hoechst+/PI+) cells were automatically calculated from the images using the Biotek proprietary software (Gen5 3.0). 

### 4.5. Caspase-1 Inflammasome Assay

The activity of caspase-1 was assayed using the Caspase-Glo^®^ 1 Inflammasome Assay (Promega, Madison, WI, USA). In this assay, Z-WEHD-aminoluciferin is degraded by caspase-1, releasing a substrate for luciferase. HCLE cells were grown in white, opaque-walled, 24-well tissue culture plates with a clear bottom, and treated as described above. Once the treatment was finished, the cells were rinsed and switched to 100 μL of phenol-red-free DMEM. Then, 100 μL of Caspase-Glo^®^ 1 Reagent was added, following the manufacturer instructions. After 1 h of incubation at room temperature, luminescence was measured in a Biotek Synergy H1 microplate reader.

### 4.6. Caspase-3 Assay

The activity of caspase-3 was analyzed using NucView^®^ 488 Caspase-3 Substrate (Biotium, Fremont, CA, USA). This substrate consists of a fluorogenic DNA dye coupled to the caspase-3 DEVD recognition sequence. After cleavage by caspase-3, the DNA dye is released and migrates to the nucleus, providing green fluorescence staining. Cells grown in 96-wells, and treated as specified, were rinsed in phenol-red-free DMEM and then incubated in 50 μL of 5 μM NucView^®^ 488 Caspase-3 Substrate in the same color-free DMEM. After incubating for 30 min at 37 °C, in 5% CO_2_, cells were observed under a Biotek Lionheart FX automated microscope. The fluorescence intensity per cell was automatically calculated from the images using the Biotek proprietary software (Gen5 3.0).

### 4.7. Transduction of UPR Sensors

To monitor the activation of UPR pathways at a single cell level, we used IRE1 and PERK sensors previously built and described [65]. As described by the authors, the PERK activity sensor contains an *ATF4* 5′ UTR region DNA sequence, where ORF1 and ORF2 are used as translation initiation in non-stressful conditions. When PERK is activated and phosphorylates eIF2a, the latter triggers translation initiation at ORF3 under ER stress, allowing the expression of the fluorescent mScarlet-I with a c-myc NLS coding sequence. The IRE1 activity sensor presents the 410–633 nucleotide sequence of the xbp1 cDNA that contains a 26 bp intron that is spliced in response to ER stress by IRE1 RNase activity. The removal of this intron leads to a frame shift that triggers translation of mNeonGreen fused with the c-myc NLS sequence.

For a higher efficiency in the delivery of the constructs, we used viral particles. Cells (3 × 10^6^ HEK293T/17 (ATC)) were plated in T75 flasks precoated with poly-L-lysine (ref) and transfected with pUMVC (MLVgag-pol), pCMV-VSG-G, and either pLHCX-ATF4 mScarlet NLS or pLHCX-XBP1 mNeonGreen NLS plasmids. A total of 9 μg of DNA was transfected (3 μg of each plasmid), using Lipofectamine 3000 (Thermo Fisher Scientific). Cell medium was collected after 48 and 72 h, and viral particles were concentrated using a Lenti Concentrator (Origene, Rockville, MD, USA). The viral titer was determined in cultures of HEK293T/17 cells using 96-well plates and serial dilution of virus for transduction.

HCLE were plated in monolayers in 96-well plates at 5000 cells/well. On the next day, the cells were transduced with retrovirus carrying either the pLHCX-ATF4 mScarlet NLS or pLHCX-XBP1 mNeonGreen NLS plasmid, at a multiplicity of infection (MOI) of 10. The cells were incubated with the retroviral particles overnight at 37 °C, in 5% CO_2_, after which the cells were changed to complete KSFM and incubated for two more days before performing the experiment.

The transduced cells were subjected to HOS and ER stress and treated with dynasore as described above. For experiments with the ATF4 reporter, PERK inhibitor GSK2606414 (Sigma-Aldrich, St Louis, MO, USA) was included in specific wells at 500 nM. Cells were then placed under the Biotek Lionheart FX automated microscope and monitored for 16 h, with the temperature controlled (37 °C) and 5% CO_2_. The expression of fluorescent particles was analyzed by automatic calculation of fluorescent-positive cells from the images, using the Biotek proprietary software (Gen5 3.0).

### 4.8. Quantitative Polymerase Chain Reaction (qPCR)

For gene expression analysis, RNA was isolated using a GeneJET RNA purification kit (Thermo Fisher Scientific) following the manufacturer’s instructions. DNA contamination was removed from columns with a PureLink^®^ DNase set (Invitrogen, Carlsbad, CA, USA). Reverse transcription was performed with a High Capacity Reverse Transcription Kit (Applied Biosystems, Foster City, CA, USA) to synthesize the first-strand cDNA from 1 μg of total RNA.

The qPCR reaction was performed using SYBR^®^ Green reagents (iTaq Universal SYBR Green Supermix; Bio-Rad, Hercules, CA, USA) with specific primers (Table 1). The following parameters were used: 30 s at 95 °C, followed by 40 cycles of 5 s at 95 °C, and 30 s at 60 °C. All samples were normalized to RNA levels of the housekeeping gene β-actin (*ACTB*) (Table 1). The comparative CT method was used for relative quantitation [66], selecting the relative amount in control cells as the calibrator.

### 4.9. Western Blot

Western blotting was performed to determine the relative phosphorylation of eIF2α, a subunit of the 126 kDa multimer eIF2. The complex dissociates when dissolved in SDS sample buffer, releasing the 36 kDa eIF2α subunit.

Cells were lysed in RIPA buffer (VWR Scientific, Franklin, MA, USA) containing a protease and phosphatase inhibitor cocktail (Thermo Fisher Scientific) and harvested using a cell scraper. The lysates were then centrifuged at 14,000× *g* for 15 min at 4 °C. Supernatants were collected and the protein concentration was assessed using a BioTek Synergy H1 microplate reader.

Protein extracts (60 μg per sample) were loaded on SDS-PAGE gels (Bio-Rad) under reducing conditions. Membranes were blocked in TBST buffer (TBST with 0.1% Tween) containing 5% BSA for 1 h at room temperature, and then incubated with anti-eIF2α, anti-P-eIF2α (both at 1:1000; Cat. No. 5324S and Cat. No. 3398S, respectively; Cell Signaling Technologies, Danvers, MA, USA), or anti-ACTB (1:4000; Cat. No. 12262; Cell Signaling) overnight at 4 °C. The blots were then incubated with IR dyes-conjugated secondary antibodies (Li-Cor Biosciences, Lincoln, NE, USA) for 1 h at room temperature. The membranes were visualized using a Li-Cor Odyssey imaging system (Li-Cor Biosciences, Lincoln, NE, USA), and the density of the different bands was assessed with the Image Studio™ acquisition software (Li-Cor Biosciences).

### 4.10. Statistical Analysis

Statistical analysis was performed using GraphPad Prism 7 (GraphPad Software, San Diego, CA, USA). The Kolmogorov–Smirnov test was used to assess the normality of data distribution, and Bartlett’s test for the homogeneity of the variances. Based on normality of the data distribution and the homogeneity of the variances, analysis of variance (ANOVA) with Bonferroni’s post-hoc test, or the Kruskal–Wallis test with Dunn’s post-hoc test, was applied for comparison. When comparing two groups, a Student’s *t*-test or Mann–Whitney *U* test were used, attending to normality of the data distribution. *p* values < 0.05 were considered statistically significant.

## Figures and Tables

**Figure 1 ijms-24-04754-f001:**
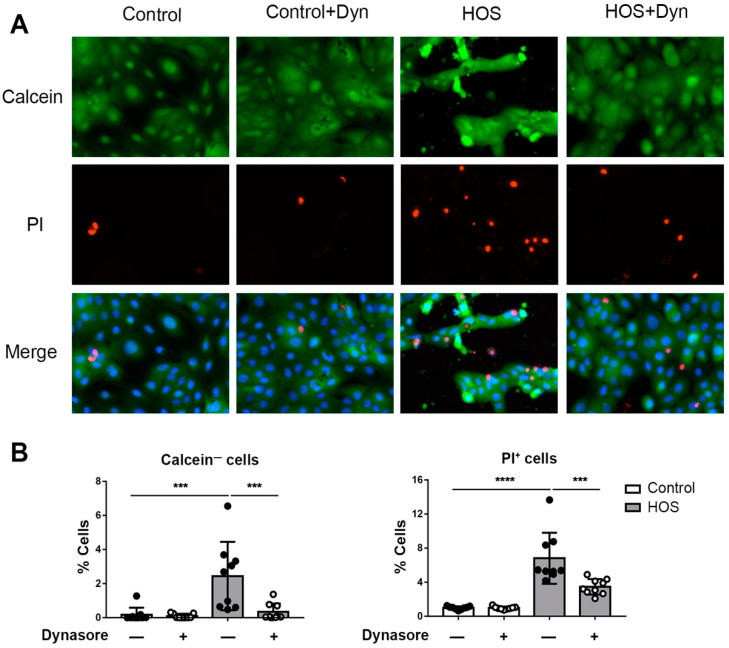
Dynasore protects against HOS-induced cell death in monolayer cultures of HCLE cells. HCLE monolayers were either left unstressed by continued incubation in serum-free cell culture medium (control), or were subjected to hyperosmolar stress (HOS), by increasing the NaCl concentration of the medium to 69 mM. Then, dynasore was added to the culture medium to a final concentration of 40 uM, while a parallel set of cultures was treated with vehicle alone. After treatment for 24 h, cells were stained with calcein-AM and propidium iodide (PI) to test the viability of the cells. Hoechst stain was used to identify all cells. (**A**) Representative images of calcein-AM (green) and PI (red) staining, with merged images also including Hoechst stain (blue). (**B**) Quantification of calcein-AM-negative and PI-positive cells. The data are presented as mean ± standard deviation. *** *p* < 0.001; **** *p* < 0.0001.

**Figure 2 ijms-24-04754-f002:**
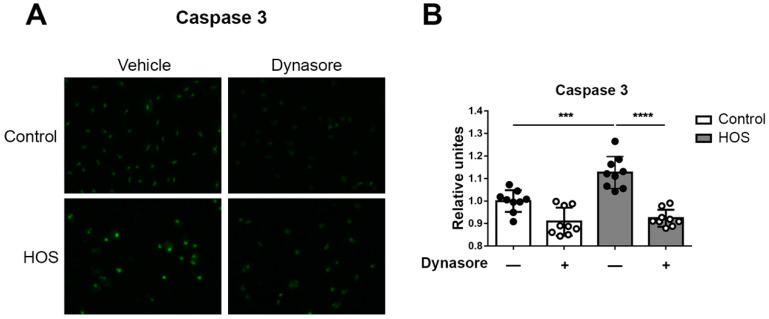
Dynasore protects against HOS-induced caspase-3 activation in monolayer cultures of HCLE cells. HCLE monolayers were either left unstressed by continued incubation in serum-free cell culture medium (control) or were subjected to hyperosmolar stress (HOS), by increasing the NaCl concentration of the medium to 69 mM. Then dynasore was added to the culture medium to a final concentration of 40 uM, while a parallel set of cultures was treated with vehicle alone. After treatment for 24 h, cells were stained with 5 μM NucView^®^ 488 Caspase-3 Substrate. (**A**) Representative images of stained cells. (**B**) Fluorescence intensity of NucView^®^ 488 Caspase-3 Substrate, showing caspase-3 activity. The data are presented as mean ± standard deviation. *** *p* < 0.001; **** *p* < 0.0001.

**Figure 3 ijms-24-04754-f003:**
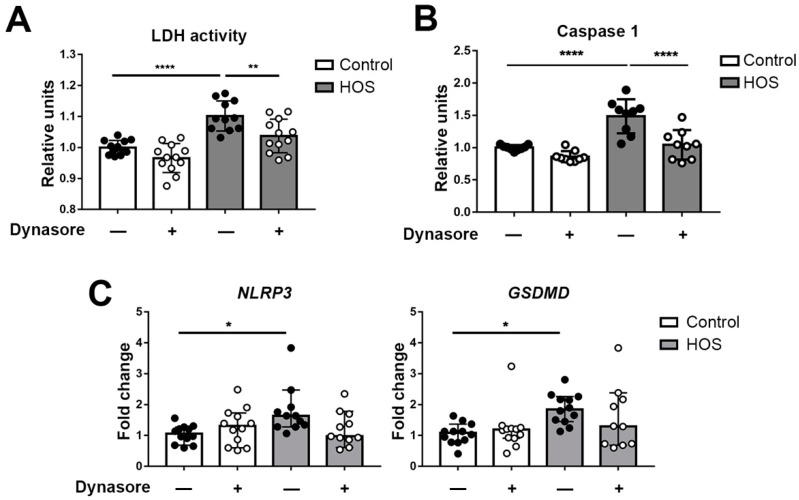
**Dynasore reduces HOS-induced cell death and activation of the inflammasome.** Stratified HCLE cells were either incubated in normal DMEM or in hyperosmolar media, by increasing DMEM NaCl concentration to 69 mM. Then, 40 μM dynasore, or the same amount of vehicle, was added to the control and hyperosmolar media as treatment. Twenty-four hours after treatment, the cell medium was collected for LDH assay, and cells were processed for either caspase-1 assay or RNA isolation and qPCR analysis. (**A**) LDH activity in cell medium was assessed by absorbance reading in a plate reader. (**B**) Caspase-1 activity results of luminescence readings after Z-WEHD-aminoluciferin cleavage and release of the luciferase substrate. (**C**) Relative gene expression of *NLRP3* and *GSDMD* was calculated with the 2^− ΔΔCt^ method, using the levels of *ACTB* expression as housekeeping and the expression in vehicle-treated control cells as the calibrator. The data are presented as mean ± standard deviation. * *p* < 0.05; ** *p* < 0.01; **** *p* < 0.0001.

**Figure 4 ijms-24-04754-f004:**
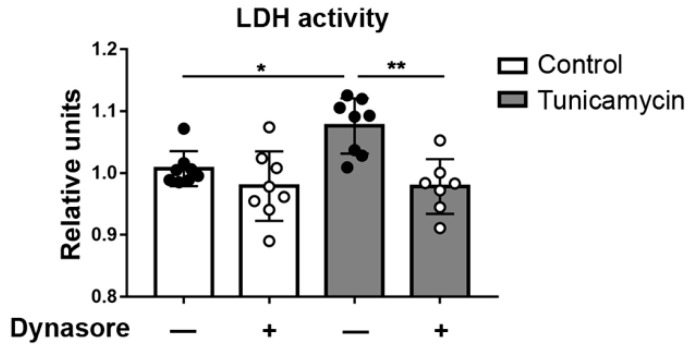
Dynasore reduces ER-stress-induced cell death. Stratified HCLE cells were incubated in DMEM with 10 μg/mL tunicamycin and 40 μM dynasore for 24 h. Equal amounts of vehicle were used in control cells and cells without dynasore. Cell medium was collected for LDH assay. The data are presented as mean ± standard deviation. * *p* < 0.05; ** *p* < 0.01.

**Figure 5 ijms-24-04754-f005:**
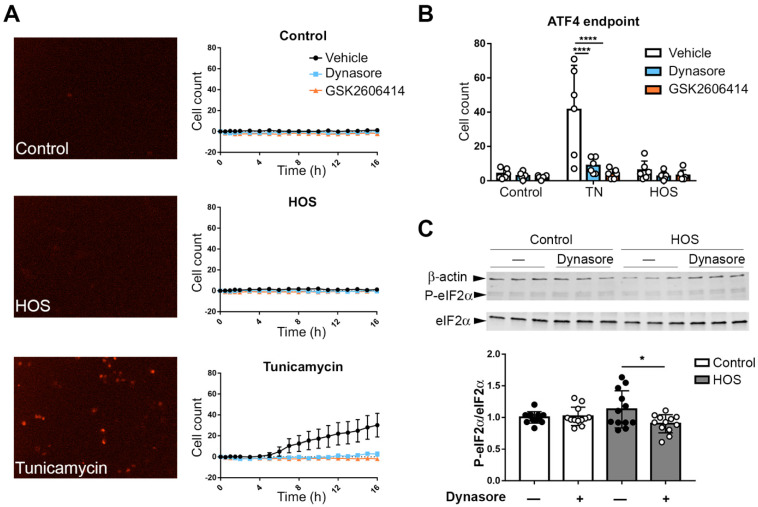
Dynasore reduces PERK activation in ER stress, but this branch is not activated by HOS. (**A**,**B**) Monolayer cultures of HCLE cells were transduced with pLHCX-ATF4 mScarlet NLS plasmid to express an ATF4 reporter. Then, 72 h later, cells were either incubated in normal DMEM (Control), DMEM with 10 μg/mL tunicamycin or in hyperosmolar media (HOS), by increasing DMEM NaCl concentration to 69 mM. Then, 40 μM dynasore, 500 nM GSK2606414, or the same amount of vehicle was added to the media as treatment for 24 h. (**A**) Cells were monitored for 16 h with a fluorescent microscope to detect the expression of ATF4 reporter. Analysis of the number of ATF4+ cells at each time point is shown. (**B**) Analysis of the number of ATF4+ cells per condition at the end of the experiment is shown. (**C**) Stratified HCLE cells were incubated in normal DMEM (control) or HOS conditions for 24 h; then, total protein was isolated. Representative images of Western blot for P- eIF2α, eIF2α, and β-actin are shown, together with densitometry analysis of the P-eIF2α/eIF2α ratio normalized to β-actin levels as a loading control. The data are presented as mean ± standard deviation. * *p* < 0.05, **** *p* < 0.0001.

**Figure 6 ijms-24-04754-f006:**
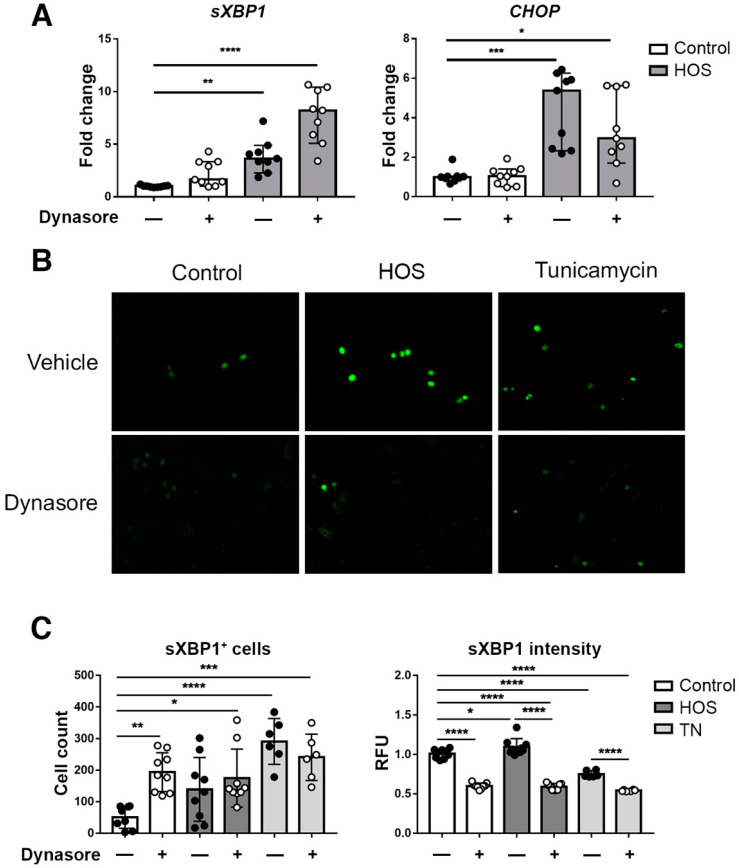
Dynasore increases the number of *sXBP1^+^*-expressing cells, but reduces its levels per cell. (**A**) Stratified HCLE cells were incubated in normal DMEM (Control) or in hyperosmolar media (HOS) by increasing DMEM NaCl concentration to 69 mM for 24 h; then, total RNA was isolated. Relative gene expression of *sXBP1* and *CHOP* was calculated with the 2^−ΔΔCt^ method, using the levels of *ACTB* expression as housekeeping and the expression in control, untreated cells as the calibrator. (**B**,**C**) Monolayer cultures of HCLE cells were transduced with pLHCX-XBP1 mNeonGreen NLS plasmid to express a sXBP1 reporter. Then, 72 h later, cells were either incubated in normal DMEM (Control), DMEM with 10 μg/mL tunicamycin or HOS conditions. Then, 40 μM dynasore, or the same amount of vehicle, was added to the media as treatment for 24 h. (**B**) Representative images of sXBP1 reporter fluorescence on each condition. (**C**) Analysis of the number of sXBP1^+^ cells per condition and sXBP1 intensity per cell are shown. The data are presented as mean ± standard deviation. * *p* < 0.05, ** *p* < 0.01, *** *p* < 0.001, **** *p* < 0.0001.

**Table 1 ijms-24-04754-t001:** Primer sequences for qPCR.

Gene	Primer Sequence
*sXBP1*	Forward: CTGAGTCCGAATCAGGTGCAG Reverse: ATCCATGGGGAGATGTTCTGG
*CHOP*	Forward: AGAACCAGGAAACGGAAACAGA Reverse: TCTCCTTCATGCGCTGCTTT
*NLRP3*	Forward: GAATGCTTGGGAGACTCAG Reverse: AGATTCTGATTAGTGCTGAGTACC
*GSDMD*	Forward: GAACTGAGTGTGGACAGAGC Reverse: CTGAGGCCAGTATGCTGAAG
*ACTB*	Forward: GTCATTCCAAATATGAGATGCGT Reverse: GCTATCACCTCCCCTGTGTG

## Data Availability

All data is contained in the figures of this paper.

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
