# Peer review of "Dynasore Protects Corneal Epithelial Cells Subjected to Hyperosmolar Stress in an In Vitro Model of Dry Eye Epitheliopathy"

_ijms, 2023, doi:10.3390/ijms24054754_

Round 1
Reviewer 1 Report
This is a well-done study on the topic that dynasore inhibits the cell death pathway triggered by hyperosmolar stress, protecting against ER stress and maintaining a homeostatic level of UPR activity driven by the UPR IRE1 branch. However, there are some points to be clarified.
- In the title, it is necessary to more clearly indicate the nature of the experiment in which this study was conducted. The current title is as confusing as the results of an animal study. I recommend the title as "Dynasore Protects Cultivated Corneal Epithelial Cells Subjected to Hyperosmolar Stress in an in vitro Model of Dry Eye Epitheliopathy."
- It would be nice if the research results and pathways were summarized graphically.
- There are some inconsistent findings and explanations in Figure 6 and related descriptions. 1) For HOS group for sXBP1, Figure 6A and Figure 6C (cell count x intensity) shows some discrepant results.
2) The expression “but dynasore treatment actually increased sXBP1 expression (Figure 6A).” in line 201 should be reconsidered because there is no statistical difference between dynasore (-) vs dynasore(+) in both control and HOS groups according to Figure 6A sXBP1 panel.
3) The expression “Therefore, although dynasore induced sXBP1 expression in more cells,” in line 212 should be reconsidered because the number of sXBP1+ cells did not differ significantly between dynasore (-) vs dynasore(+) in both the HOS and TN groups, which might be expected to have an effect by dynasore (only significantly different in control group).
Author Response
This is a well-done study on the topic that dynasore inhibits the cell death pathway triggered by hyperosmolar stress, protecting against ER stress and maintaining a homeostatic level of UPR activity driven by the UPR IRE1 branch. However, there are some points to be clarified.
COMMENT: In the title, it is necessary to more clearly indicate the nature of the experiment in which this study was conducted. The current title is as confusing as the results of an animal study. I recommend the title as "Dynasore Protects Cultivated Corneal Epithelial Cells Subjected to Hyperosmolar Stress in an in vitro Model of Dry Eye Epitheliopathy."
RESPONSE: Thank you, we agree that it would be best to further clarify and have added in Vitro to the title.
COMMENT: It would be nice if the research results and pathways were summarized graphically.
RESPONSE: We included a graphical abstract with our paper in order to clearly define the pathways examined and the results we obtained. Perhaps the reviewer did not have the opportunity to review this? We believe this should serve the proposed purpose well.
COMMENT: There are some inconsistent findings and explanations in Figure 6 and related descriptions. 1) For HOS group for sXBP1, Figure 6A and Figure 6C (cell count x intensity) shows some discrepant results.
2) The expression “but dynasore treatment actually increased sXBP1 expression (Figure 6A).” in line 201 should be reconsidered because there is no statistical difference between dynasore (-) vs dynasore(+) in both control and HOS groups according to Figure 6A sXBP1 panel.
3) The expression “Therefore, although dynasore induced sXBP1 expression in more cells,” in line 212 should be reconsidered because the number of sXBP1+ cells did not differ significantly between dynasore (-) vs dynasore(+) in both the HOS and TN groups, which might be expected to have an effect by dynasore (only significantly different in control group).
RESPONSE: Thank you for your comments. We have tried to clarify these points:
1) Although dynasore+HOS treated cells showed a trend towards increase in sXBP1, the differences with HOS alone were not significant as the reviewer notes in the point 2. It is also necessary to note that Figures 6A and 6C are measuring things that are different at various levels. First, Fig.6A shows results on stratified cells while Fig.6C was performed in monolayer cells (as this was the best way to monitor the expression of the reporter). The stratified cultures contain a higher amount of cells, with different subtypes compared to monolayer cultures (e.g., apical cells) and they can show slight differences in their response to stress. Second, Fig.6A shows qPCR results for sXBP1, while Fig.6C is a reporter assay. PCR can be more accurate when quantifying the number of transcripts than a visual reporter assay and is more sensitive to detect slight variations.
2) We agree. We have modified this sentence as “An increase in sXBP1 was observed in cells subjected to HOS alone that was not reverted in dynasore-treated cells, which actually presented a trend toward more increased sXBP1 expression (Figure 6A)”
3) We have modified the sentence as ”Therefore, although dynasore did not reduce the number of cells expressing sXBP1,”
Reviewer 2 Report
The work “Dynasore Protects Corneal Epithelial Cells Subjected to Hyperosmolar Stress in a Model of Dry Eye Epitheliopathy” is an excellent paper evaluating the use of dynasore in protecting corneal epithelial cells under HOS.
The authors make a strong case for the asset as a protecting molecule by showing that it basically inhibits many programmed cell death that are commonly triggered by HOS, such as cell death, caspase 3 activation, inflammasome activation, UPR activation. Even when PCR results for sXBP1 and CHOP didn’t show a reduction with dynasore, the authors conducted single cell level analysis with sXBP1 reporter and came to interesting conclusions. Unfortunately the authors do not measure MMP-9 in any way (although pathways measured can be related to its expression), but it seems that this is that osmolarity and MMP-9 are the only commercially available essays that could potentially measure the variables of interest in this study. The introduction, results, graphs and methods are excellent and this paper should be approved pending minor revisions
Line 35: currently more dry eye disease drugs have been approved, including lotheprednol (kala) and varenicline (Tyrvaya). Please edit
Line 37: “and neither was effective against epitheliopathy in human clinical trials “. This is deceiving, as some of the trials actually showed decreased punctate epitheliopathy, including for both lifitegrast and cyclosporine as they have shown decreased rates of corneal staining in some of their trials. Please revise
Discussion: Can you briefly elaborate on potential speculations of efficacy (which biomarkers to test) and possible limitations when applying this drug to humans?
